# Xanthommatin is Behind the Antioxidant Activity of the Skin of *Dosidicus gigas*

**DOI:** 10.3390/molecules24193420

**Published:** 2019-09-20

**Authors:** Jesús Enrique Chan-Higuera, Hisila del Carmen Santacruz-Ortega, Ángel A. Carbonell-Barrachina, Armando Burgos-Hernández, Rosario Maribel Robles-Sánchez, Susana Gabriela Cruz-Ramírez, Josafat Marina Ezquerra-Brauer

**Affiliations:** 1Departamento de Investigación y Posgrado en Alimentos; University of Sonora (UNISON), 83000 Hermosillo, Sonora, Mexico; jeen.chhi@gmail.com (J.E.C.-H.); armando.burgos@unison.mx (A.B.-H.); rsanchez@guayacan.uson.mx (R.M.R.-S.); 2Departamento de Investigación en Polimeros y Materiales, University of Sonora (UNISON), 83000 Hermosillo, Sonora, Mexico; hisila.santacruz@unison.mx; 3Department of Agro-Food Technology, Escuela Politécnica Superior de Orihuela, Universidad Miguel Hernández de Elche (UMH), Orihuela, Alicante 03312, Spain; angel.carbonell@umh.es; 4Unidad Hermosillo, Universidad Estatal de Sonora (UES); 83100 Hermosillo, Sonora, Mexico; susycr13@hotmail.com

**Keywords:** antioxidant activity, chromatography, ommochromes, spectroscopy, xanthommatin

## Abstract

Marine bioactive compounds have been found in very different sources and exert a very vast array of activities. Squid skin, normally considered a discard, is a source of bioactive compounds such as pigments. Recovering these compounds is a potential means of valorizing seafood byproducts. Until now, the structure and molecular properties of the bioactive pigments in jumbo squid skin (JSS) have not been established. In this study, methanol–HCl (1%) pigment extracts from JSS were fractionated by open column chromatography and grouped by thin-layer chromatography in order to isolate antioxidant pigments. Antioxidant activity was determined by the 2,2-diphenyl-1-picrylhydrazyl (DPPH^●^) and 2,2′-azino-bis-(3-ethylbenzothiazoline-6-sulfonic acid) (ABTS^●+^) radical scavenging assays and ferric reducing power (FRAP) assay. Fractions 11–34 were separated and grouped according to flow rate values (F1–F8). Fractions F1, F3, and F7 had the lowest IC_50_ against ABTS^●+^ per milligram, and fractions F3 and F7 showed the lowest IC_50_ in the FRAP assay. Finally, fraction F7 had the highest DPPH^●^ scavenging activity. The chemical structure of the F7 fraction was characterized by infrared spectroscopy, ^1^H nuclear magnetic resonance, and electrospray ionization–mass spectrometry. One of the compounds identified in the fraction was xanthommatin (11-(3-amino-3-carboxypropanoyl)-1-hydroxy-5-oxo-5H-pyrido[3,2-a]phenoxazine-3-carboxylic acid) and their derivatives (hydro- and dihydroxanthommatin). The results show that JSS pigments contain ommochrome molecules like xanthommatin, to which the antioxidant activity can be attributed.

## 1. Introduction

Marine bioactive compounds show noteworthy and natural properties that support their nutraceutical and pharmaceutical potential and are regarded as more secure options in contrast to synthetic drugs and food additives. Marine bioactive compounds have been isolated and characterized from different sources, like plants, microorganisms, algae and animals (vertebrates and invertebrates) [1]. Among the different invertebrates studied, one of the most notable for its commercial impact and tonnage of capture is the jumbo squid (*Dosidicus gigas*) [2]. As with most marine species, only the squid muscle is of economic importance. Obtaining clean squid filet requires the removal of skin and other anatomical regions. This process creates waste that accounts for over 40% of the total squid weight [3].

Squid skin (normally considered a discard) is particularly rich in biologically active compounds, such as gelatin, collagen, and their peptides, as well as pigments [4,5,6,7]. The pigments found in jumbo squid skin are a part of its defense mechanism, which has been perfected through years of evolution. They can instantaneously change their coloration to adapt to the environment. This unique combination of neuromuscular organs present on their skin is formed by an elastic saccule that allows the chromatophores to expand and relax, producing different colors [8]. The pigments in cephalopods have been previously characterized as ommochromes, and they constitute a class of polycyclic aromatic compounds that are synthesized through the metabolic pathway of tryptophan oxidation [9]. Their basic structure is a ring of fenoxazone (ommatins) or phenothiazine (ommins and possibly ommidins). Among the ommatins found in invertebrates, xanthommatin and their derivatives like hydro- and dihydroxanthommatin have been related to biological activities, mainly the oxidative protection against free radicals [10].

Ommochromes can act as antioxidants, and their antioxidative mechanisms can be achieved through chelating activity, and they can also act as primary antioxidants by scavenging radicals such as singlet oxygen and superoxide anions [11]. Ommochromes prevent photodamaging effects in the eyes of marine species [12]. Ethanolic pigment extracts obtained from jumbo squid skin (*Dosidicus gigas*) were used as antioxidants against the heat-induced rancidity of cod liver oil [7]. Jumbo squid pigments have also been proven to exert antimicrobial activity in iced mackerel and hake by inhibiting trimethylamine, microbial proteolysis, and lipolysis [13,14].

Although the antioxidant activity of squid skin extracts has been examined in some studies, most reports have described antioxidant proteins and peptides [15]. Moreover, no reports exist on the identification of the pigments responsible for the antioxidant activity of this important fishery resource. The aim of this work was to isolate and identify the pigments responsible for the antioxidant activity detected in squid skin (*Dosidicus gigas*). 

## 2. Results

### 2.1. Isolation of the Bioactive Pigments 

The liquid-phase column to open column method resulted in the elution of 34 fractions from the raw extract. The obtained fractions were analyzed according to their physical characteristics, as well as the results of solubility tests (data not shown). Fractions 1–10 were excluded from further analysis because no compounds were collected, as determined by the equal weight of the vial before and after evaporating the solvent. The remaining fractions (11–34) were analyzed by identifying their separation pattern in thin-layer chromatography plates. From the obtained results, the fractions were grouped according to the number of bands in each extract, as well as their Rf values. They were reclassified for a total of eight fractions, designated F1–F8.

### 2.2. Antioxidant Activity

IC*_50_* values showed that the compounds in fractions F1, F3, and F7 had the ability to perform the single electron transference (SET) mechanisms against the 2,2′-azino-bis-(3-ethylbenzothiazoline-6-sulfonic acid (ABTS*^●+^*) radical. In the ferric reducing power ferric reducing power (FRAP) method, the highest electron transference activity was detected in fractions F3 and F7. Lastly, the hydrogen atom transference (HAT) capacity was the highest (*p* < 0.05) in fraction F7, against the 2,2-diphenyl-1-picrylhydrazyl (DPPH*^●^*) radical. Although the three techniques evaluate the ability to stabilize different radical species, the F7 fraction showed the highest activity in all of them (Table 1). From this information, it was decided to proceed with the chemical characterization of fraction F7.

### 2.3. Structure Elucidation

FT-IR, ^1^H NMR and electrospray ionization–mass spectrometry (ESI-MS) spectra were compared against previously published data and spectrum databases. 

Figure 1 presents the IR spectrum of fraction F7. The peak associated with the stretching of primary amines is observed at 3298 cm*^−^*^1^. This signal is also associated with the flexure of the primary amine, which is detected as a peak at 705 cm*^−^*^1^. In the region of 3250–3600 cm*^−^*^1^, a characteristic peak of the –OH functional group is observed, which overlaps with the previously described amino group. The presence of aromatic rings is associated with signals between 3000 and 3300 cm*^−^*^1^, which are related to aryl carbons. This is corroborated by the signals located between 1600 and 2000 cm*^−^*^1^, which are related to aromatic overtones. 

One of the main compounds present in fraction F7 was established by comparing its ^1^H NMR spectrum with previously published data [16,17,18,19]. The main signals detected by ^1^H NMR (CD_3_OD, 400 MHz) were δ 8.40 (s, 1H), 8.01 (d, 1H, *J* = 8.0 Hz), 7.96 (t, 1H, *J* = 8 Hz), 7.92 (s, 1H), 7.45 (d, 1H, *J* = 8Hz), 3.91 (dd 1H, *J* =5.03 Hz, 5.52 Hz), 3.56(dd *J* = 5.03 Hz, 1.92 Hz), and 3.52 ppm (dd, 1H, *J* = 5.52 Hz, 1.92Hz) (Figure 2). Moreover, the ^1^H NMR spectrum shows signals due to sp3 carbons at δ 3.87 (Figure 2, letters f and g; m, 2H), aromatic protons at δ 7.88 (Figure 2, letters d and e; d, 2H), and amine group protons at δ 7.70 (s, 1H). 

### 2.4. Electrospray Ionization–Mass Spectrometry

The positive ESI-MS exhibited a quasimolecular peak at m/z 424 (M + H)+ in full scan mode (Figure 3). Thus, it was inferred that the relative molecular weight of the compounds found were about 423 to 427.

## 3. Discussion

The separation of the extract was achieved using an open column, taking advantage of the characteristics of the pigments in squid skin that have been previously reported [20,21]. A high affinity between the sample and the extract was observed. Additional observations include a delayed elution and more effective recovery as the solvent polarity increased. This behavior could be due to the chemical structure of the silica gel, which contains a large proportion of hydroxyl groups. Ommochromes have hydrophobic parts and polar groups (amino and hydroxyl, particularly), and the latter can form hydrogen bonds and interact strongly with silica. Previous studies have reported that compounds with polar functional groups can be separated using a polar stationary phase, even if there are strong intermolecular interactions [22].

Since oxidation reactions do not all follow a single mechanism, evaluating antioxidant capacity through several assays is widely encouraged to allow the assessment of different modes of antioxidant action. The DPPH^•^, ABTS^+^, and FRAP methods were used to evaluate the electron transfer (SET) and hydrogen atom transfer (HAT) mechanisms of antioxidant activity. The results obtained in the antioxidant part of the study were used to identify the fraction with the highest activity by both mechanisms and thus characterize the compounds responsible for this biological activity. While fractions F1 and F3 showed the lowest IC_50_ values along with F7, fraction F7 had the highest in the DPPH method, and this information indicated that the compound or compounds in this fraction could either donate electrons or hydrogen atoms. The results obtained in this study strongly suggest that the pigments in fraction F7 are able to exert such mechanisms, as suggested by previous in silico studies [23]. These findings are relevant because of the importance of the hydrogen atom transference mechanism is relation to the prevention of peroxidation reactions in foodstuffs, as well as oxidation of biologically important molecules [5]. One of the ommochromes was identified as a potent electron and hydrogen donor, namely, xanthommatin [11-(3-amino-3-carboxypropanoyl)-1-hydroxy-5-oxo-5H-pyrido [3,2-a]phenoxazine-3-carboxylic acid]. The signals of two other similar molecules were also observed: hydro- and dihydroxanthommatin. The structures of rhodommatin, ommatin D, hydroxykynurenine, and xanthommatin have functional groups that are related to antioxidant action, primarily hydroxyl linked to aromatic rings.

Structure elucidation is a complex process that involves the interpretation and comparison of different assays, in order to establish the presence of certain molecules. IR, ^1^H NMR, and ESI-MS are techniques regarded as useful tools that help in the identification of molecules present in natural extracts. Previous reports have successfully elucidated the structure of pigments present in natural sources [24,25]. The identification of the functional groups in the molecules was achieved through IR analysis. The signals of certain functional groups associated with both antioxidant activity and compounds in the ommochrome family were detected. The amino group, both primary and secondary, in fraction F7 can act as an antioxidant given its electron transference capacity. A cyclic amine is present in the structure of the compound [26]. The tendency to donate electrons is related to the fact that the amine concentrates its electronic density in the aromatic ring. In addition, the amine forms stable resonance structures with the aromatic ring, which is absent once the amine is protonated. However, the peak attributed to the –OH functional group overlaps with that of the amino group. The characteristics of the sample, combined with the results of other techniques described later, suggest the presence of these groups. The antioxidant capacity of the hydroxyl groups has been widely reported, and phenolic compounds are recognized as being some of the most potent antioxidants in nature. The mechanism is driven by the resonance stabilization of the aromatic ring [25]. The IR results suggest that, in effect, aromatic rings are present in the compounds in fraction F7. In general, ommochromes have a basic structure of phenoxazone, which is derived from the amino acid tryptophan [27]. In addition to these results, Aubourg et al. [7] reported that a peak at 1740 cm^−1^ is characteristic of xanthommatin, an ommochrome present in squid skin extracts obtained with ethanol/acetic acid. Moreover, the data obtained for fraction F7 agree with previous reports on xanthommatin [28].

The ^1^H NMR spectrum of fraction F7 shows signals that can be attributed to the presence of a phenoxazone core [17]. This kind of compound has been previously detected in the skin of some cephalopods [16,17]. The NMR spectrum, along with the FT-IR spectrum, confirms the presence of functional groups associated with antioxidant activity. 

It has been established that phenoxazone cyclizes to dehydroxanthommatin, which oxidizes itself to xanthommatin (molecular weight, 423 g mol^−1^) [18]. The ion at *m/z* 427 was assumed to be the corresponding quasimolecular ion of another ommochrome, such as dihydroxanthommatin; whereas hydroxanthommatin has a molecular weight of 425 g mol^−1^ [19]. Therefore, from the FT-IR and IR results, combined with the ESI-MS results, the presence of ommochromes, such as xanthommatin in fraction F7 is supported.

The compounds that showed radical scavenging activity and ferric reduction antioxidant power in *Dosidicus gigas* skin extracts are ommochromes. In this study, the ommochromes xanthommatin, hydroxanthommatin, and dihydroxanthommatin were found as molecular components responsible for the antioxidant activity of the extract, and its antioxidative mechanisms have been described as hydrogen atom transference and single electron transference. The results of this study confirm the presence of ommochromes with biological activity in jumbo squid extracts. This information can help establish that jumbo squid skin pigments have potential use in the food industry as a preventive agent against oxidation. Currently, there is an ongoing study on the application of the fraction with the greatest antioxidant activity for the preservation of a food product and its possible toxicological risk, as well as the contribution of antioxidant activity of the different compounds present in the fraction. 

## 4. Materials and Methods 

### 4.1. Sample Preparation

Jumbo squid (*Dosidicus gigas*) was obtained from a local establishment in Hermosillo, México (29°05′56″N, 110°57′15″W) and immediately skinned. About 10 kg of fresh skin was frozen at −80 °C, freeze-dried (LabConco, Kansas City, MO, USA), and grinded. Samples were kept at −20 °C until further analyses were performed.

### 4.2. Pigment Extraction

Freeze-dried skin was mixed with acidified methanol (1% HCl; 1:20 *w/v* proportion) and sonicated for 5 min. Samples were centrifuged (10,000× *g* for 15 min), the supernatant was collected, and the extraction solvent was removed using a rotary evaporator (R-100, Büchi, Switzerland). 

### 4.3. Fractioning by Open Column Chromatography

The raw squid skin extract was fractionated using the liquid-phase column to open column technique. Silica gel with a particle size of ≤0.063 mm (Merck, Darmstadt, Germany) was placed as a stationary phase in a glass column, and a series of solvent combinations (all of analytical grade) were used as the mobile phase; this information is shown in Table 2.

### 4.4. Thin-Layer Chromatography

The compounds in the previously obtained fractions were preliminarily identified through thin-layer chromatography (TLC). Static glass plates coated with silica gel were used as the stationary phase, and a combination of methanol/ethyl acetate/ammonium hydroxide (75:25:5) was used as the mobile phase. The samples were injected (10 μL fraction) and allowed to run for 30 min in a chamber saturated with solvents. The rate of flow (Rf) of the bands was observed and calculated to regroup those exhibiting the same pattern of separation.

### 4.5. Antioxidant Activity

The in vitro antioxidant activity of the collected fractions was evaluated by the three spectrophotometric assays. 

2,2-Diphenyl-1-picrylhydrazyl (DPPH) radical scavenging activity was the first assay employed to determine the antioxidant activity, according to the method of Brand-Williams et al. [29]. Aliquots of each collected fraction (1 mg mL^−1^) were dissolved in 1 mL of methanol, followed by the addition of 4 mL of a DPPH solution (0.004% w/v) in methanol. The samples were placed at 25 °C for 30 min, and the absorbance was read at 517 nm. 

The second assay was the 2,2′-azino-bis-(3-ethylbenzothiazoline-6-sulfonic acid) (ABTS) radical scavenging [30]. The ABTS radical cation (ABTS^●+^) was activated by adding 7 mmol ABTS in water and 0.14 mmol potassium persulfate. The mixture was incubated in the dark at 25 °C for 16 h. After mixing the ABTS^●+^ solution (Abs_734_nm = 0.70) with the samples, the mixtures were incubated for 30 min. The absorbance was recorded at 734 nm (Cary 50 UV–Vis, Agilent Technologies, Toluca, Mex., Mexico). 

The third analysis involved the ferric reducing or antioxidant power of the samples [31]. An aliquot of 100 μL of the samples (1 mg mL^−1^) was mixed with 1 mL of FRAP reagent (10 mM tripyridyl triazine prepared in 40 mM HCl, 25 mL acetate buffer, and 2.5 mL of 20 mM FeCl_3_H_2_O), and the reaction mixture was incubated at 25 °C for 30 min. The absorbance increase was registered at 593 nm (Cary 50 UV–Vis, Agilent Technologies). 

The concentration of the sample required to scavenge 50% of the DPPH^●^ and ABTS^●+^ radicals, or to reduce 50% of the Fe^3+^ atoms was determined using an inhibition curve using different concentrations of the fractions.

### 4.6. Spectroscopic Methods

The infrared spectrum of the sample was obtained with a Perkin Elmer spectrometer (Frontier MIR/FIR, Waltham, Massachusetts, USA). An attenuated total reflectance (ATR) technique was performed. The spectra were collected at 25 °C between 4000 and 400 cm^−1^, accumulating 30 scans per spectrum. A blank spectrum was recorded to exclude any cross-contamination. The spectrum was expressed in wavenumber (cm^−1^) versus transmittance percentage.

The ^1^H NMR spectrum of the fraction was obtained on a Bruker Avance 400 nuclear magnetic resonance spectrometer operating at 400 MHz. The sample was dissolved in a mixture of deuterated methanol (CD_3_OD) and dimethyl sulfoxide, using tetramethylsilane (TMS) as the internal reference. Chemical shifts were referenced to the solvent peaks, and the values were recorded in δ. The multiplicities of the ^1^H NMR signals are indicated as s (singlet), d (doublet), and m (multiplet). 

### 4.7. Electrospray Ionization–Mass Spectrometry

The mass spectrum of the fraction was obtained using a mass spectrometer (Agilent Technologies 6100 Quadrupole LC/MS, Santa Clara, California, USA). The dissolved sample was injected into a mixture of methanol with acetonitrile. The MS was operated in positive mode to analyze the compounds present in the squid skin extract. The data were acquired in scan mode using an m/z range of 300–650. The ESI technique was used because it is nondestructive and thus maintains the complete structure of the molecules in the fraction.

### 4.8. Statistical Analysis

Data on the antioxidant activities of isolated jumbo squid skin (JSS) pigments are reported as the average of three determinations and analyzed using analysis of variance (ANOVA) with Tukey–Kramer tests. The IC_50_ values of the fractions were obtained through a linear regression analysis. 

## Figures and Tables

**Figure 1 molecules-24-03420-f001:**
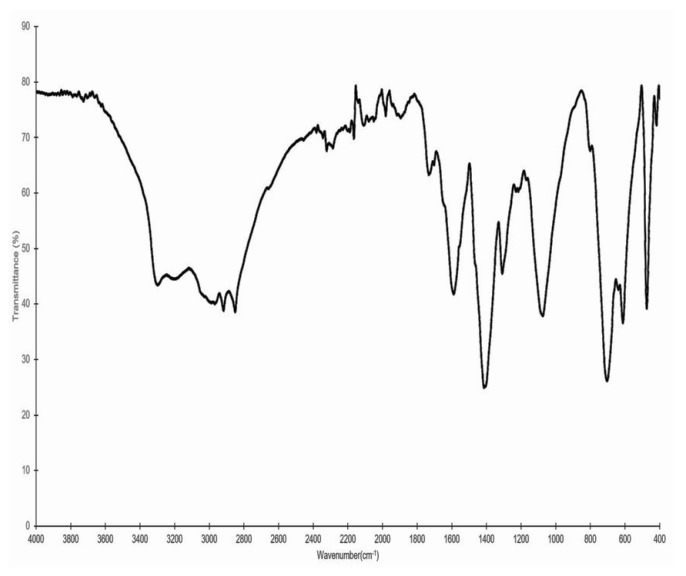
Infrared spectrum of fraction F7, which showed the highest activity in the DPPH, ABTS, and FRAP antioxidant assays.

**Figure 2 molecules-24-03420-f002:**
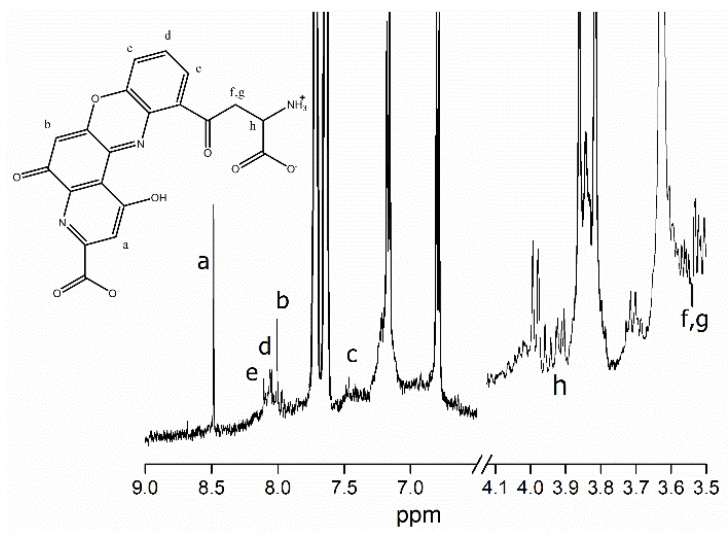
^1^H NMR spectra of fraction F7, which showed the highest activity in the DPPH, ABTS, and FRAP antioxidant assays.

**Figure 3 molecules-24-03420-f003:**
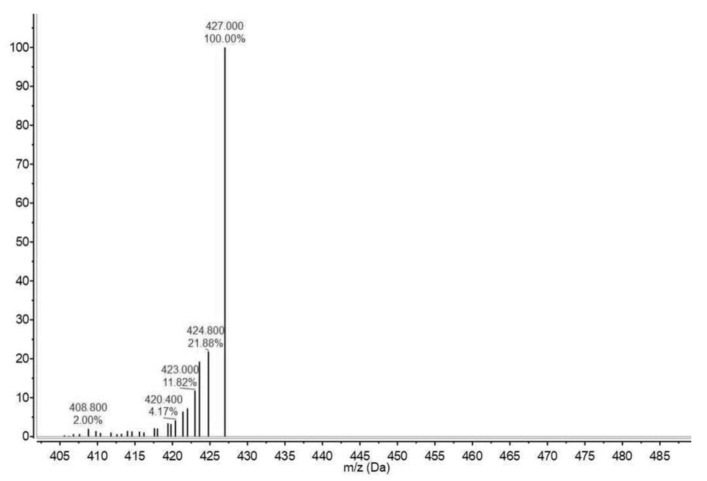
Electrospray ionization–mass spectrum of fraction F7, which showed the highest activity in the DPPH, ABTS, and FRAP antioxidant assays. The data were acquired in scan mode using an *m/z* range of 300–650.

**Table 1 molecules-24-03420-t001:** Antioxidant activity of the collected fractions of squid skin extract, evaluated by three methods.^1^

Fraction	ABTS^2^ (IC_50_ mg mL^−1^)	FRAP^2^ (IC_50_ mg mL^−1^)	DPPH^2^ (IC_50_ mg mL^−1^)
F1	2.12 ± 0.11^a^	6.54 ± 0.06^c^	6.49 ± 0.04^e^
F2	2.77 ± 0.08^c^	4.15 ± 0.11^b^	4.67 ± 0.03^c^
F3	2.07 ± 0.11^a^	2.52 ± 0.20^a^	3.56 ± 0.08^b^
F4	10.2 ± 0.02^e^	12.34 ± 0.33^d^	8.69 ± 0.09^e^
F5	6.02 ± 0.05^d^	6.89 ± 0.07^c^	9.52 ± 1.01^e^
F6	2.25 ± 0.02^b^	3.98 ± 0.10^b^	5.34 ± 0.05^d^
F7	2.08 ± 0.02^a^	2.25 ± 0.09^a^	2.60 ± 0.04^a^
F8	5.78 ± 1.02^d^	7.30 ± 0.03^c^	3.44 ± 0.09^b^

^1^ The values represent the average of three repetitions ± standard deviation. ^2^ Different letters in the same column indicate significant differences (*p < 0.05*). ABTS: the 2,2′-azino-bis-(3-ethylbenzothiazoline-6-sulfonic acid; FRAP: ferric reducing power; DPPH: 2,2-diphenyl-1-picrylhydrazyl.

**Table 2 molecules-24-03420-t002:** Solvents used as the mobile phase during open column chromatography.

Mixture of solvents	Proportion
Ethyl acetate/Methanol	60:40
Ethyl acetate/Methanol	40:60
Ethyl acetate/Methanol	20:80
Methanol	100
Acetic acid/Water	5:95
Acetic acid/Water	10:90
Ammonium hydroxide/Water	4:96
Ammonium hydroxide/Water	8:92

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
