# Peer review of "Xanthommatin is Behind the Antioxidant Activity of the Skin of Dosidicus gigas"

_molecules, 2019, doi:10.3390/molecules24193420_

Round 1
Reviewer 1 Report
This paper was too simple and it should be revised thoroughly.
Table 1 should express the IC50 of different fractions about antioxidant activities, which were better to analyze the antioxidant activities of different fractions. Figure 1 partly showed a 1H NMR spectrum for fraction 7, which was of a mixture but not a purified compound, justly provided a corresponding information with some structural moiety. How to determine the structure?3 The integral results of the proton signals in the 1H NMR spectrum is not consistent with the proton numbers. H-a was assigned as δ8.41 (d, 3H). In the structure of xanthommatin, there is only one proton, which should described as a singlet.
In the line 123 of page 4, the author described “The negative ESI-MS exhibited a quasimolecular peak at m/z 424 [M + H]+in full scan mode(Figure 3)” .Firstly , the main spectroscopic peak was 427, but not 424, this is inaccurate. In addition, this is not in a negative mode. According to the mass data at m/z 424 [M + H]+ , the molecular weight should be 423. Furthermore, ESI-MS data is insufficient to determine molecular weight. The 1H NMR, IR and ESI-MS provide a limited evidence for structural characterization. The author should add the chromatography of Fraction 7 to analysis the purity of xanthommatin, and then provide the 13H NMR, and HRESI-MS data. It was better to evaluate the compounds of other fractions such as F3. And further analyze the relation of the compounds and structures.Author Response
Dear reviewer, thank you for your kind observations and suggestions.
Please see the attachment.

Reviewer 2 Report
I am pleased to review the assigned manuscript. Overall, a it looks very good to me. Authors really did a wonderful job and presented very nice & relevant literature. I have few suggestions that can be incorporated into revised version.
Abstract looks good but just wondering if it can cover the whole theme I suggested some of the latest references, it would improve content and the novelty of the research. Authors can incorporate the related research carried out at the University of Queensland, Australia.
Marine-based species comprising approximately one half of the total universal biodiversity, the oceans and aquatic environment in general offer a plenty of resource for novel bioactive components. Marine species comprises bioactive compounds and much attention has been paid to them as they play pivotal role in disease prevention and maintenance of human health. These marine bioactive compounds exhibit significant biological properties that contribute to their nutraceutical and pharmaceutical potential and are also considered to be safer alternatives to some existing synthetic drugs. These bioactivities include anti-oxidant, anti-thrombotic, anti-coagulant, anti-inflammatory, anti-proliferative, anti-hypertensive, anti-diabetic and cardio-protection activities, making them attractive nutraceuticals and pharmaceutical compounds.
Results
To be honest, I didn't find the enough justification of LC-ESI-QTOF/MS, please check this section.
In general, manuscript looks wonderful to and authors did a wonderful job.
Author Response
Dear reviewer, thank you for your kind observations and suggestions.
Please see the attachment.

Round 2
Reviewer 1 Report
ok
Author Response
We are very glad you appreciated our work. We were pleased to attend to your reviews and comments.
Thank you.